# Do digital innovations for HIV and sexually transmitted infections work? Results from a systematic review (1996-2017)

Jana Daher,[1] Rohit Vijh,[1] Blake Linthwaite,[1] Sailly Dave,[1] John Kim,[2] Keertan Dheda,[3] Trevor Peter,[4] Nitika Pant Pai[1,5]

[1]Division of Clinical Epidemiology, Research Institute of the McGill University Health Centre, Montreal, Canada
[2]National HIV/AIDS Labs, National Labs, Winnipeg, Manitoba, Canada
[3]Department of Pulmonology, UCT Lung Institute, University of Cape Town, Cape Town, South Africa
[4]Clinton Health Access Initiative (CHAI), Boston, USA
[5]Department of Medicine, McGill University, Montreal, Quebec, Canada

**Correspondence to**
Dr Nitika Pant Pai;
Nitika.Pai@mcgill.ca

## ABSTRACT

**Objective** Digital innovations with internet/mobile phones offer a potential cost-saving solution for overburdened health systems with high service delivery costs to improve efficiency of HIV/STI (sexually transmitted infections) control initiatives. However, their overall evidence has not yet been appraised. We evaluated the feasibility and impact of all digital innovations for all HIV/STIs.

**Design** Systematic review.

**Setting/participants** All settings/all participants.

**Intervention** We classified digital innovations into (1) mobile health-based (mHealth: SMS (short message service)/phone calls), (2) internet-based mobile and/or electronic health (mHealth/eHealth: social media, avatar-guided computer programs, websites, mobile applications, streamed soap opera videos) and (3) combined innovations (included both SMS/phone calls and internet-based mHealth/eHealth).

**Primary and secondary outcome measures** Feasibility, acceptability, impact.

**Methods** We searched databases MEDLINE via PubMed, Embase, Cochrane CENTRAL and Web of Science, abstracted data, explored heterogeneity, performed a random effects subgroup analysis.

**Results** We reviewed 99 studies, 63 (64%) were from America/Europe, 36 (36%) from Africa/Asia; 79% (79/99) were clinical trials; 84% (83/99) evaluated impact. Of innovations, mHealth based: 70% (69/99); internet based: 21% (21/99); combined: 9% (9/99). All digital innovations were highly accepted (26/31; 84%), and feasible (20/31; 65%). Regarding impacted measures, mHealth-based innovations (SMS) significantly improved antiretroviral therapy (ART) adherence (pooled OR=2.15(95%CI: 1.18 to 3.91)) and clinic attendance rates (pooled OR=1.76(95%CI: 1.28, 2.42)); internet-based innovations improved clinic attendance (6/6), ART adherence (4/4), self-care (1/1), while reducing risk (5/5); combined innovations increased clinic attendance, ART adherence, partner notifications and self-care. Confounding (68%) and selection bias (66%) were observed in observational studies and attrition bias in 31% of clinical trials.

**Conclusion** Digital innovations were acceptable, feasible and generated impact. A trend towards the use of internet-based and combined (internet and mobile) innovations was noted. Large scale-up studies of high quality, with new integrated impact metrics, and cost-effectiveness are

## Strength and limitations of this study

► An updated and comprehensive systematic review/meta-analysis of all innovations in HIV/STI.
► Evaluation of study quality with biases, subgroup analyses and sensitivity analyses.
► Evaluation of metrics and measures for objective and subjective data.
► Limited data were reported from Sub-Saharan Africa and Southeast Asia (29%, 29/99).
► Limited evidence (18/99, 18%) was available for STIs (other than HIV).
► Limited data on cost-effectiveness from high burden settings.
► A lack of integrated online impact metrics to evaluate internet-based eHealth innovations.

needed. Findings will appeal to all stakeholders in the HIV/STI global initiatives space.

## INTRODUCTION

HIV/STIs remain a public health concern worldwide—a million new HIV/STIs are acquired every day, with cumulative disease burden exceeding 500 million infections.[1–5] Regarding HIV, countries are working hard to achieve the new UNAIDS 90-90-90 treatment targets[6]; however, structural and societal barriers, such as stigma, low socioeconomic status and geographical isolation, impede access to quality care for marginalised populations who are disproportionately impacted by the HIV/AIDS epidemic.[7 8] Likewise, a lack of timely testing and poor retention in care impairs the efforts to control HIV/STIs.[7 9 10] To improve early testing, linkage and retention in care, healthcare systems globally are seeking solutions to improve population engagement, awareness and education, and efficient care for their hard-to-reach populations. It is imperative to plug gaps in healthcare service delivery.[11 12] Digital innovations

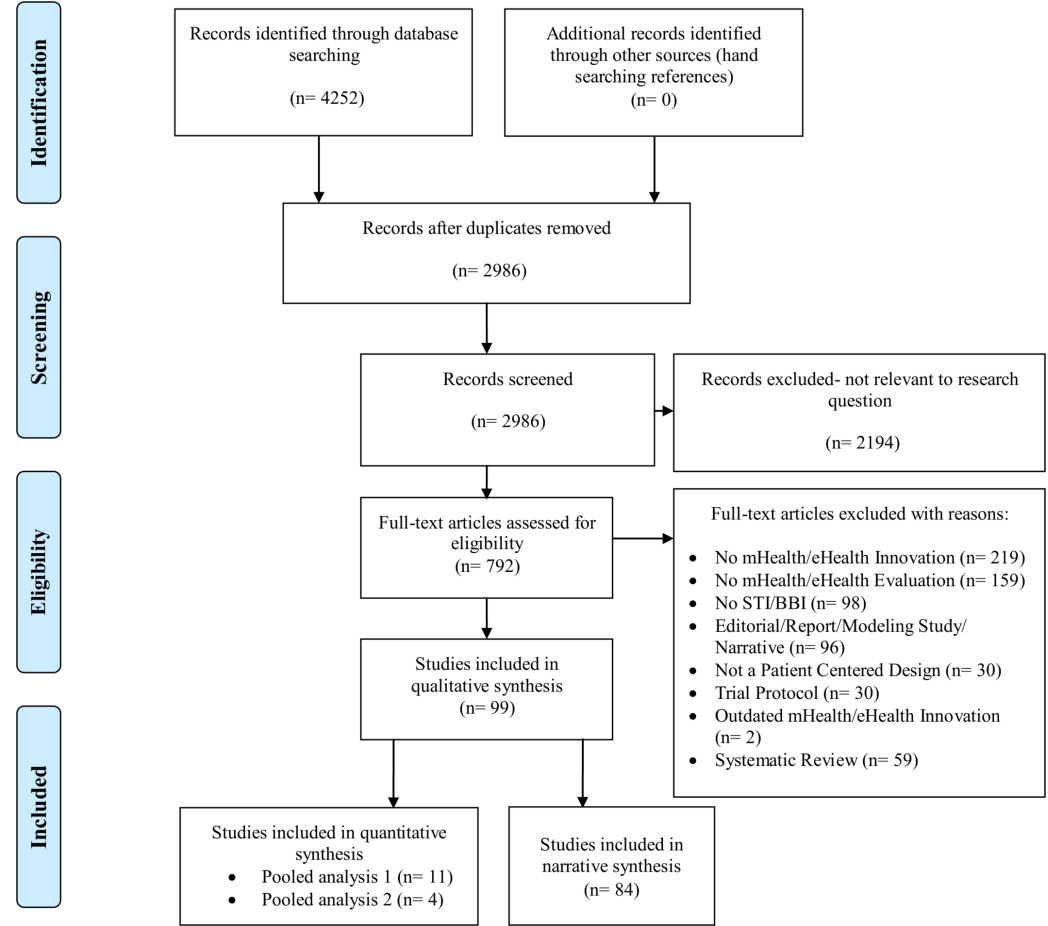

**Figure 1** PRISMA (Preferred Reporting Items for Systematic Reviews and Meta-Analyses) flow diagram.

such as electronic health (eHealth), mobile health (mHealth) and combined innovations offer promising solutions to improve health service delivery. eHealth encompasses non-internet and internet-enabled mHealth as well as other internet-based health interventions. These innovations, together with expanded mobile and internet networks, global connectivity and affordability, present opportunities to change the future landscape of healthcare service delivery.

The World Bank estimates that globally 96% of the world's population and 70% of the world's poorest have access to a mobile phone.[13] Of seven billion, two billion (30%) individuals own a smartphone; approximately 50% of mobile phone users access the internet through their phones.[14][15] Technological access has created a portal for social media and other internet-based health interventions.[16] A rapid diffusion of mobile phones and internet technologies are prime drivers of this disruptive phenomenon in health, aptly titled, the creative destruction of medicine.[17] In recent years, visionary foundations (*Grameen, Bill and Melinda Gates Foundation, UNAIDS, Vodafone*) have, with funding, created opportunities for innovative thinking in health. To date, 95 countries have evaluated some digital health innovations.[11] This is most evident in under-resourced settings where low-cost and

sustainable solutions are needed to solve complex global health challenges.[18]

Digital innovations were first used in non-communicable diseases and later became popular in infectious disease.[19] In the field of HIV/STIs, a *Lancet* study demonstrated the effectiveness of mHealth-based short message service (SMS) innovations on adherence to antiretroviral therapy (ART).[20] As novel digital innovations and strategies continue to be developed and tested, many smaller reviews and systematic reviews were published. However, a vast majority of these reviews only evaluated a single innovation (eg, SMS, social media), one or two outcomes and restricted exploration in select subgroups (people living with HIV (PLHIV), pregnant women, adolescents, men who have sex with men (MSM)).[21–27] These reviews failed to provide a comprehensive summary of all innovations for programme planning and research. Due to a rapid expansion of digital innovations, and an increased popularity of combined innovations (2013), a need for a comprehensive up-to-date synthesis on all innovations for HIV/STIs was felt.

Our primary objective was to generate a high-quality overview/systematic review that summarizes all digital innovations across all populations and outcomes in HIV/STIs. Our secondary objective was to inform researchers,

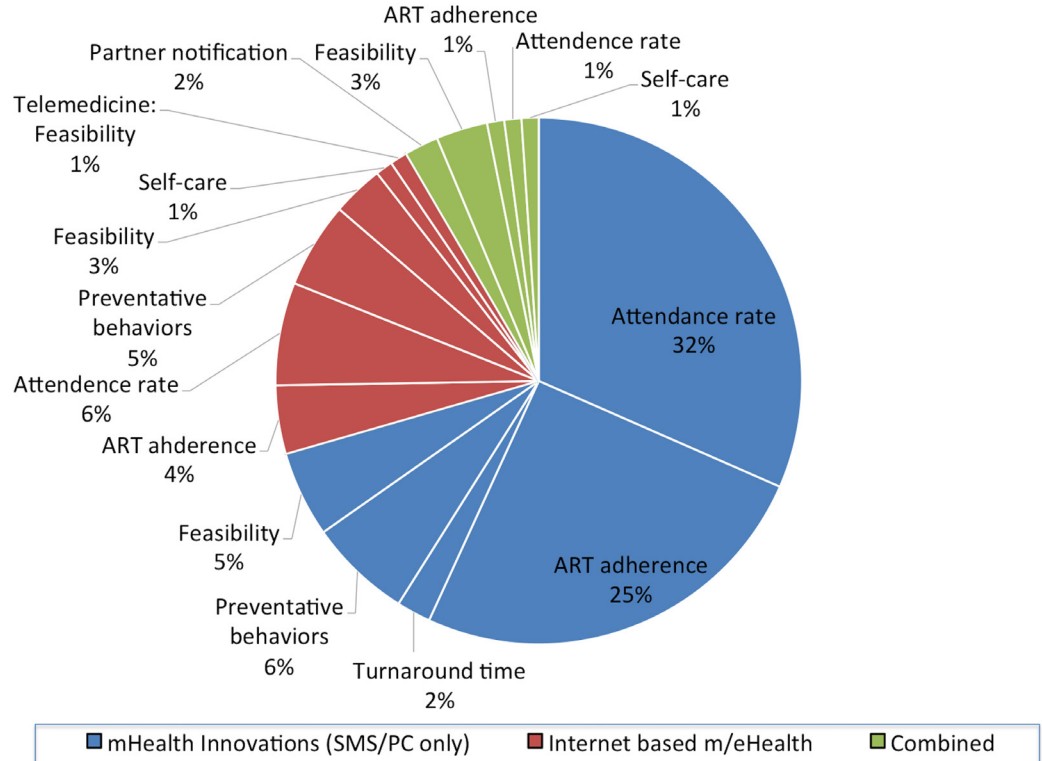

**All Innovations by Outcome Type**

**Figure 2** All innovations by outcome type (font size enlarged). ART, antiretroviral therapy; SMS, short message service

policy makers and funders with evidence for their decisions on implementation and scale-up.[11]

## METHODS
PRISMA (Preferred Reporting Items for Systematic Reviews and Meta-Analyses) and Cochrane guidelines were followed.[28]

### Data sources and searches
We searched MEDLINE via PubMed, Embase, Cochrane CENTRAL and Web of Science for a 21-year period from February 1996 up to March 2017, with no language restrictions.

### Search strategy
Keywords used were HIV, AIDS, STI, mhealth, mobile health, ehealth, telemedicine, mobile applications and social media. For a full search strategy, refer to online supplementary appendix file 1. (#1 ('HIV' [MeSH] OR 'acquired immunodeficiency syndrome' [tiab]), OR #2 (sexually transmitted infections [mh] OR sexually transmitted disease* [tiab]), AND #3 ('mHealth' [tiab] OR 'mobile health' [tiab] OR short messag* [tiab] OR 'eHealth' [MeSH] OR 'telemedicine' [MeSH] OR social medi* [tiab] OR 'mobile applications' [tiab]).

### Study selection
Two reviewers independently screened and evaluated citations for eligibility (JD and RV) and two others (BL

and SD) independently assessed quality. A senior reviewer was consulted (NPP) for discordance.

### Eligibility criteria
Any clinical trials or observational study designs that evaluated any digital (mHealth/eHealth) technology with any reported outcomes (refer to figure 1) were included.

### Data abstraction
Two reviewers (RV, JD) independently abstracted all the data. A prepiloted data abstraction form was used to abstract the following items: study design, study population, sample size, digital innovation type, HIV/STIs, outcome measures (eg, impact, acceptability and feasibility) and metrics (eg, attendance rate, completion rate, satisfaction) (refer to online supplementary appendix file 2). We referred to a previously published framework to define and further classify the following metrics for impact, acceptability and feasibility.[29]

### Subgroup pooled analyses
We classified study designs and then classified digital innovations into three groups[30]:
1. mHealth (SMS and phone calls only, that is, non-internet based);
2. internet-enabled mHealth and other internet-based eHealth (mobile application, website, online cam-

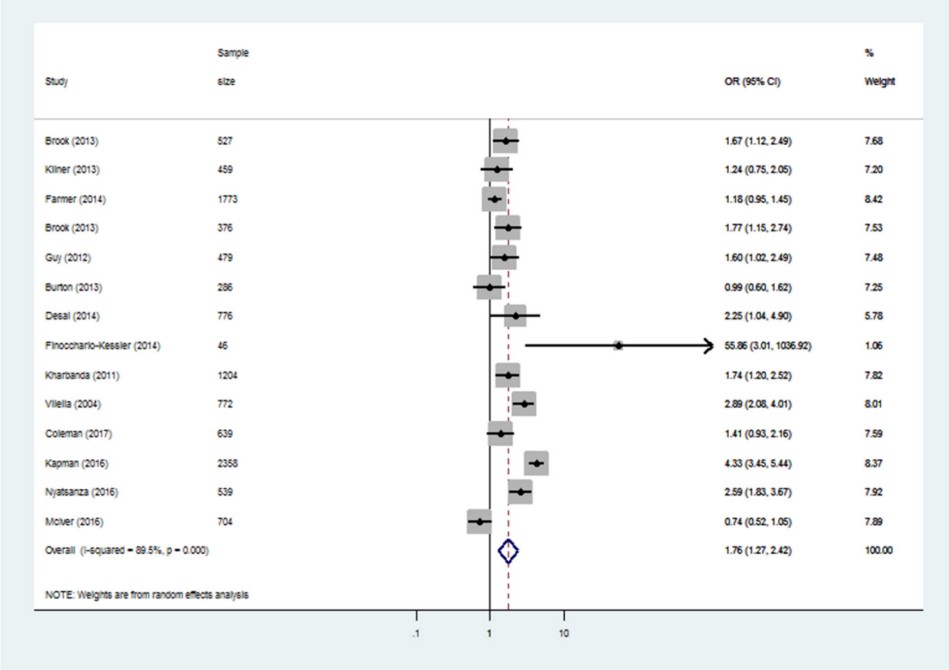

A

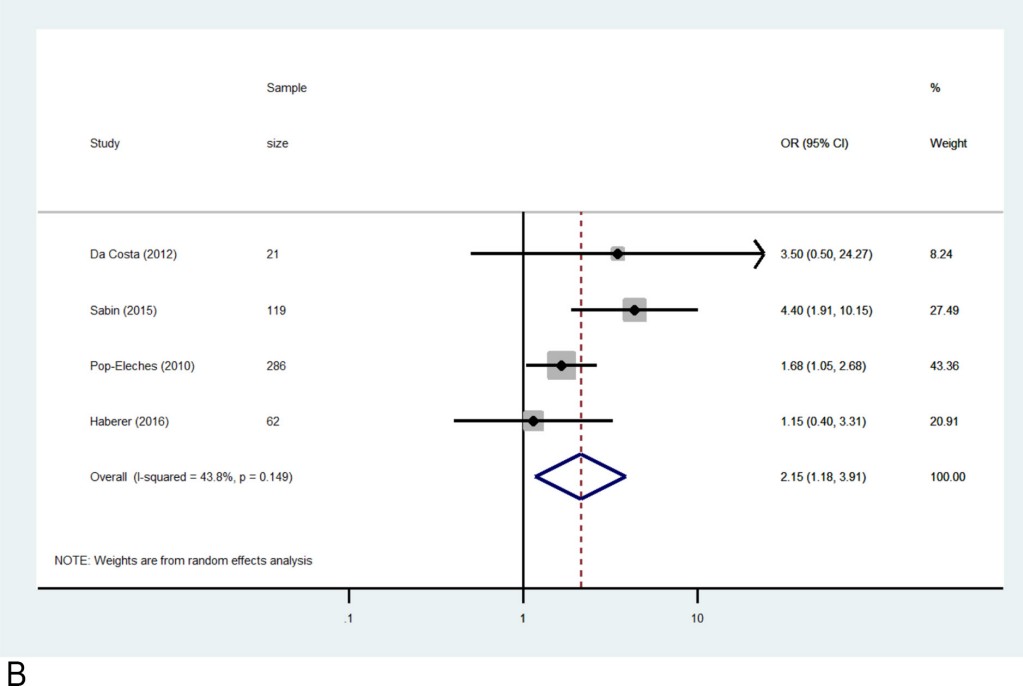

B

**Figure 3** Subgroup analyses.

paign, streamed soap opera videos, avatar-guided computer programs);
3. combined innovations (innovations that combined both mHealth (SMS/phone calls) with internet enabled mHealth/eHealth).

Only one subgroup reported similar outcomes which could be pooled, SMS and phone calls, for the following outcomes: (1) clinic attendance with SMS and (2) ART adherence via Medication Event Monitoring System (MEMS) caps, with SMS. We pooled these outcomes using a random effects subgroup analysis. Given the diversity in the sample populations between studies, we used the random effect meta-analysis model with the DerSimonian and Laird estimator (moments method) of the between-study variance to calculate the pooled effect. We generated forest plots for visual representation of heterogeneity and pooled OR with 95% CI. We

performed all statistical analyses using Stata/IC, V.13 (Stata).[31]

## Narrative analysis

We narratively described all other data using as follows:

Digital innovations were classified into the following groups based on the strength of evidence: high/strong evidence (metrics at 75%–100%), moderate evidence (51%–74%) and low/weak evidence (50% or less).

## Acceptability

Acceptability was defined as the receptivity in using digital innovations.

## Feasibility

Feasibility was defined as the perceived convenience in using digital innovations. It was reported with various metrics: completion, retention, response and referral rates.

## Impact

Impact was defined as a statistically significant improvement in measured outcomes compared with a comparator group (ie, control group or baseline observations). The metrics used to evaluate impact were (1) attendance rate, (2) ART adherence, (3) risk reduction, (4) self-care and (5) partner notification. Impact measures were evaluated on two criteria: effect size and precision. Effect size was assessed when data on a comparator group were made available. Precision of the effect estimate was assessed whenever reported, as it reflects the variance or spread of results.

## Quality assessment

We assessed study quality for both clinical trials and observational studies. We used the Cochrane Risk of Bias Tool for trials, and the Newcastle-Ottawa quality assessment scale for observational studies.

## RESULTS

Of 4252 citations identified through our extensive search, 792 were selected for full-text screening, and 99 citations met our inclusion criteria and were included in this review for evidence synthesis (refer figure 1).

## Study characteristics

By geographical location, 37% (37/99) of studies were conducted in North America, 26% (26/99) in Sub-Saharan Africa, 24% (24/99) in Europe, 7% (7/99) in Oceania, 3% (3/99) in South-East Asia and 2% (2/99) in South America.

By study design, the majority were trials: 38% (38/99) were RCTs, 16% (16/99) uncontrolled trials and 1% (1/99) non-randomised controlled trials. Others included quasi-experimental studies, of which many used historical controls (24%, 24/99) and observational studies (ie, cross-sectional and feasibility studies) (20%, 20/99).

HIV was the most frequently reported infection (74%, 73/99 studies), followed by chlamydia/gonorrhoea (CT/GC) (10%, 10/99). Combinations of HIV with STIs (eg, syphilis) (8%, 8/99), human papillomavirus (HPV) (4%, 4/99) and hepatitis A/B/C (HBV) (4%, 4/99) were also reported.

In terms of study populations, PLHIV were prominent across studies (42%, 42/99) followed by other high-risk groups (ie, MSM/bisexual men, drug users, pregnant women/mother–infant pairs, African-Americans, sex workers and visible minorities) (28%, 28/99), general clinic attendees (16%, 16/99), CT/HBV-infected individuals (4%, 4/99) and residents of a specific area (9%, 9/99).

## Innovations

Digital innovations were documented across the spectrum.

mHealth innovations (SMS/phone calls only) were evaluated in 70% (69/99) of studies.[20 32–99] Seventy-two per cent (50/69) were SMS-based and 28% (19/69) used phone calls or a combination of both (refer to figure 2 and see online supplementary appendix file 3).

Internet-enabled mHealth and other internet-based eHealth innovations were evaluated in 21% (21/99) of studies.[100–120] These innovations consisted of many different forms: social media and online campaigns (9/21), avatar-guided computer programs (2/21), mobile applications (5/21), combination of social media and websites (2/21), websites (1/21), telemedicine services (1/21) and streamed soap opera videos (1/21) (refer to figure 2 and see online supplementary appendix file 3).

Combined innovations were evaluated in 9% (9/99) of studies.[121–129] Innovations consisted of SMS+websites/interactive websites (4/9), SMS+mobile application (3/9) and SMS+social media (including online campaigns) (2/9) (refer to figure 2 and see online Supplementary appendix file 3).

## Measures and metrics

A vast majority (84%, 83/99) of studies focused on impact measure and metrics, while about 12% (12/99) focused only on feasibility and the remaining 4% (4/99) on acceptability. Within impact measures, metrics such as clinic attendance rates were reported in 45% (37/83) of studies, followed by ART adherence at 35% (29/83), HIV/STIs risk reduction behaviours at 13% (11/83), turnaround time from testing to treatment at 2% (2/83), partner notification at 2% (2/83) and self-care at 2% (2/83).

## Analyses

### Subgroup pooled analyses

It was possible to perform subgroup analyses on outcomes that were consistently documented: clinic attendance in 14 quasi-experimental studies that used SMS reminders and MEMS-based ART adherence in four randomised controlled trials (RCTs) evaluating SMS. The pooled estimate for the impact of SMS reminders on attendance

rates was 1.76 (95% CI: 1.28 to 2.42) (refer to figure 3A). The pooled estimate for the impact of SMS on ART adherence tracked via MEMS caps was also significant, OR=2.15 (95% CI: 1.18 to 3.91) (refer to figure 3B).[32 47 48]

### Narrative analysis
#### Impact
##### Non-internet-based mHealth (SMS/PC only)
Of 69 studies, positive results were reported for the following outcomes: clinic attendance (63%, 19/30 studies, of which 84% reached statistical significance), ART adherence (63%, 15/24 studies, of which 93% reached statistical significance), turnaround time from testing to treatment (67%, 2/3 studies). However, SMS reported a limited effect on risk reduction behaviours (3/7, 43%).

##### Internet-based mHealth/eHealth
Studies evaluating internet-based eHealth innovations (21/99) reported results that were largely in favour of the following innovations: social media-based interventions for clinic attendance; avatar-guided and mobile applications for ART adherence; social media, avatar and soap opera videos for risk reduction behaviours; mobile app for self-care.

Social media contributed to higher testing uptake rates in all studies (6/6, 100%). A social media-based campaign increased HIV testing by 252% (n=1500; 19% from baseline 5.4%, p<0.01) and Syphilis testing by 248% (18.8% from baseline 5·4%, p<0.01), while another campaign increased HIV testing by 52% compared with control (n=625; 63.7% vs 42% in controls, OR=2.9 (95% CI: 1.8 to 4.7)).[100 115] Four campaigns guaranteed rapid in-home HIV testing for all those who requested it online.[100 101 108 111 116]

Avatar-guided programs and mobile applications improved ART adherence in all studies (4/4). Statistically significant outcomes were reported in 2/4 programs (50%). These were (1) a personalised avatar-guided computer program improved adherence (n=240; p=0.046); (2) a mobile application with immunosuppression graphs and medication reminders lowered viral load (n=28; p=0.023) and improved adherence (p=0.03) as well.[102 104] In the other two studies, an avatar-guided program improved viral suppression and a mobile application allowed for 100% adherence, but these were underpowered to detect a significant effect (n=76 and n=28, respectively).[107 110]

Social media, avatar and soap opera videos were successful at reducing risky sexual behaviour in all the reported studies (5/5). However, significant results were reported in only three of five studies: (1) social media-based interventions decreased unprotected sex acts by 65% (n=31; 3.11 vs baseline 8.96, p=0.042); (2) soap opera videos on HIV prevention reduced condomless sex by 78% (n=117; 78% reduction from baseline, p<0.001)[103 106]; (3) an avatar-guided computer program also lowered the odds of HIV transmission (n=240;

OR=0.46, p=0.012).[102 103 106] Even in two underpowered studies, social media-based interventions led to 40% and 67% higher condom uptake (n=70 and n=50, respectively).[105 117]

A mobile application increased self-care in the sole study in this category (1/1). A significantly higher self-care performance among chronic HBV-infected individuals was reported compared with controls (n=53; p=0.001).[112]

##### Combined innovations
Studies evaluating combined innovations (9/99) showed success of social media+SMS in increasing clinic attendance and partner notification; interactive websites+SMS in improving ART adherence; and mobile app+SMS in increasing self-care. Among the five impact studies, 80% reported a favourable outcome. An online campaign with SMS services increased CT, GC and HIV tests uptake by 41%, 91% and 190%, respectively[123]; an interactive website with SMS reminders improved ART adherence in drug users (n=20; p=0.02)[121]; a social media-based partner notification with SMS increased notified contacts by 144% (63.5% in 2011 from baseline 26% in 2010)[126]; and a mobile app with SMS significantly improved self-care performance in HIV-infected individuals compared with baseline (n=19; p=0.002).[129]

### Acceptability and feasibility
Overall, across studies that assessed acceptability/feasibility, digital innovations were found to be highly acceptable and feasible (75%–100%)%) in 26/31 and 20/31 studies, respectively. mHealth innovations (SMS/PC only) were highly acceptable and feasible in 81% (13/16) and 75% (12/16) of studies, respectively.

Internet-based mHealth/eHealth innovations were highly acceptable and feasible in 92% (11/12) and 45% (5/11) of studies, respectively. All included innovations (ie, avatar, mobile app, social media and streamed videos) were highly acceptable.[102–104 106 107] While avatar-guided program was rated high on feasibility, social media-based strategies were found to be less feasible[101–103]

Combined innovations were highly acceptable and feasible in 67% (2/3) and 75% (3/4) of studies, respectively.[121 124] The innovations that were rated high involved a combination of SMS and interactive websites.

### Quality
Studies were individually evaluated on quality criteria, and biases were noted. Across trials, losses to follow-up were reported in 31% of RCTs and 55% of quasi-trials. Additionally, biases (ie, misclassification, recall bias) were of concern in 58% of the RCTs and 64% of quasi-randomised trials (refer to online supplementary appendix file 4 and 5).

In observational studies, confounding (68%) and selection bias (66%) were observed (refer to online supplementary appendix file 6). Studies with small sample sizes, low power or insufficient follow-up time (eg, 3 weeks

or less) sometimes provided contradictory results when objective and subjective metrics evaluated the same outcome.

## DISCUSSION
### Summary of findings
Overall, digital innovations reported positive effects on key metrics. We noted a strong positive effect of digital innovations on clinic attendance rates (70%; 26/37), ART adherence (69%; 20/29), risk reduction behaviours (67%; 8/12) and self-care (100%; 2/2). SMS/phone calls were not able to reduce risky sexual behaviours; however, social media-based interventions, particularly interactive social media, were effective in reducing risky sexual behaviours. Acceptability was found to be high for all innovations. Feasibility estimates also remained high for all innovations, except for social media-based interventions, possibly due to a perceived lack of privacy and confidentiality. Combined innovations may thus offer promise in plugging this feasibility gap, with internet-based innovations compensating for limitations in SMS-only strategies and vice versa.

While mHealth (SMS/phone calls only) innovations were highly effective in improving clinic attendance, ART adherence and turnaround time from testing to treatment, they did not report on other outcomes. It should be noted that SMS and phone calls alone failed to reduce risky sexual behaviours, which was not surprising as it is challenging to reduce risky behaviours with a prescriptive SMS alone. Population engagement is essential for risk reduction through qualitative research.

While internet-based mHealth/eHealth innovations (social media, avatar-guided computer programs, mobile apps and soap opera videos) demonstrated positive evidence on impact metrics, not all studies reached statistical significance. Those that failed to report a statistically significant improvement in ART adherence had small sample sizes and were underpowered to detect these outcomes (n=76 vs n=240), and had less frequent sessions over a shorter evaluation period (2 sessions over 6 months vs 4 sessions over 9 months).[102 107] For mobile applications, studies which reported significant effects recruited participants with varying level of adherence,[104 110] compared with studies which had high adherence at baseline (≥95%) and did not show significance (due to smaller changes in effect). For social media-based campaigns, the two studies that did not reach statistical significance in reducing risky sexual behaviours lacked an interactive component and simply displayed educational material, while the study that showed significant effect engaged the participants by allowing them to contact professional cognitive behavioural therapists via live chat sessions.[103 105 117]

In terms of quality, confounding and selection bias were noted in observational and quasi-experimental studies, and loss to follow-up in some trials. Nevertheless, the overall validity of the findings from this review was not threatened by biases, as a large proportion of our data were derived from trials. While clinical trials were generally high quality, observational studies were medium to low quality.

Consistent reporting of metrics was lacking, which prevented a comprehensive meta-analysis. Objectives, end points, metrics and measures are equally important in feasibility studies and must be well designed to generate high-quality evidence.

Our review is an exhaustive assessment of the role of digital innovations in improving prevention and care for HIV/STIs. Our findings resonate with many smaller systematic reviews, which have separately evaluated individual components of digital innovation, such as SMS-based mHealth.[22 23 130–137] Other systematic reviews evaluating social media-based interventions reported similar findings to ours, in improved testing uptake or improvements in sexual health.[25–27 138 139]

Our review makes a valuable addition to the growing body of evidence by highlighting the success of other interactive and engaging innovations such as avatar-guided computer programs, mobile apps, streamed soap opera videos and combined innovations. These integrated innovations and programs are gaining in popularity because of their power to engage rural and urban audiences at many levels.

Designing combined innovations that are complementarity of various media, methods, platforms and messaging may deliver best results. This complementarity may also encourage participant engagement to improve prevention and care metrics and measures sustainably over time. Engagement is challenging when only one innovation (eg, mHealth SMS/phone calls only) is the sole focus, where boredom is likely.

### Caveats and implications for future research
There are some caveats to considering design and evaluation of innovations. Future research needs to be focused on tailoring innovations to the context and population, and program objectives. Innovations aiming to reduce risky sexual behaviours could be interactive and tailored to the setting and population, with a deep understanding of patients' needs and preferences.[137 140 141] Any communication with patients could be customised for timing to avoid fatigue with its uptake. For example, patients may be more responsive to weekly versus daily SMS ART reminders.[32 142]

Study quality is essential to generating meaningful results. Large and representative samples of the underlying population and sound statistical techniques during data analysis or sampling methodology can minimise selection bias. Exploring reasons for differential losses to follow-up could inform future studies. Wherever possible, a control group should be included to differentiate the Hawthorne effect from the effect of the intervention.[143] Trials and impact designs can prevent or reduce confounding. Following checklists, like the one by the WHO mHealth Technical Evidence Review

Group on mHealth innovations, is suggested and encouraged.[144]

Objective measures (eg, HIV/STIs diagnosis, viral load) are desired in reporting of quantitative outcomes, over subjective self-reported data (eg, condom use, self-reported adherence). This could potentially reduce some biases (misclassification biases or desirability/recall biases) that are observed with subjective reporting.

Qualitative data are rich and complement the understanding of all the contextual and population needs, and capture the dynamics of sustainability and change. They need to be integrated with quantitative data to provide a holistic picture of uptake of any digital innovation.

Quality of digital data will merit from an improvement. Across studies, a lack of integrated online impact metrics in evaluating the success of innovations was evident. With continuously evolving digital media, inventing new ways to evaluate acceptability and feasibility becomes necessary. For example, some studies tracked online metrics via Google analytics.[74 100 101 121–124] Synergy with industry powered metrics could be a new wave to measure success of digital innovations.

To scale up proven innovations, a multistakeholder engagement is necessary. For that, data and metrics that appeal to all sections of stakeholders will be needed. In addition to improving the quality of randomised controlled trials and quasi-experimental impact studies, qualitative studies, cost-effectiveness studies and usability studies are also needed.

### Implications for policy and practice

In consonance with other systematic reviews, evidence at scale and over time was scarce.[138] This limits the projection of the long-term sustainability and cost-effectiveness of digital innovations. More evidence on scale-up, cost-savings and cost-effectiveness from Sub-Saharan Africa and Asia is needed. Future investments that incentivise both the development and evaluation of combined innovations by government and industry alike, and focus on sustainability of digital innovations with public and private partnerships, are urgently needed.

### CONCLUSION

To control HIV/STIs globally, we need novel and disruptive innovations that will uniquely impact health outcomes across the spectrum of access, engagement, treatment and retention so as to impact health service delivery. On one hand, mHealth (SMS/phone calls only) and internet-based mHealth/eHealth were found acceptable, feasible and offered complementarity in improving prevention and care of HIV/STIs. On the other hand, when combined, they provided customised and contextualised solutions for hard-to-reach populations.

Innovations need to be proven for impact and cost-effectiveness, using a combination of clinical trials, quasi-randomised studies, observational studies and qualitative research studies. Integrating these innovations across

various levels of healthcare with clear evaluation, monitoring and documentation of metrics will facilitate their integration within existing health service delivery models so as to efficiently impact health outcomes over time.

Findings from this comprehensive review will be informative to all stakeholders—innovators, researchers, healthcare practitioners, policy makers and funders—worldwide seeking evidence on integrating and funding innovations, to make (impact) the entire spectrum of HIV/STI care.

**Acknowledgements** The authors would like to acknowledge Ms. Megan Smallwood for her assistance in editing the manuscript.

**Contributors** NPP, JD: concept, design. NPP: data critiquing, write-up, critique, and overall responsibility of the data. JD: data synthesis, write-up, critiquing. RV, BL and SD: data synthesis, write"up and critique. JK, TP and KD: write"up and critique.

**Funding** Grand Challenges Canada Transition to Scale; Grant number 0710"05. FRSQ Salary Award Chercheur"Boursier Junior 2.

**Competing interests** None declared.

**Provenance and peer review** Not commissioned; externally peer reviewed.

**Data sharing statement** No additional data are available. This is a systematic review/syntheses of existing studies, therefore all data are reported in the tables.

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
