## [Reviewer comments · BMJ Open]

ARTICLE DETAILS

TITLE (PROVISIONAL)	Do Digital Innovations for HIV and Sexually Transmitted Infections work? Results from a Systematic Review (1996-2017).
AUTHORS	Daher, Jana; Vijh, Rohit; Linthwaite, Blake; Dave, Saily; Kim, John; Dheda, Keertan; Peter, Trevor; Pai, Nitika

VERSION 1 – REVIEW

REVIEWER	Claudia Estcourt Glasgow Caledonian University, UK None
REVIEW RETURNED	28-May-2017

GENERAL COMMENTS	Thank you for giving me the opportunity to review this paper. I think it is an area of current interest but I struggled with the very broad body of literature it attempted to review systematically & just felt the remit of the review was so wide that the findings were just too broad to be of much help. It was good to see an assessment of quality of the literature reviewed but in the light of clearly documented issues of bias across the greater part of the RCTs & quasi RCTs included, I felt the authors did not address this in their abstract, nor discuss the possible implications of this sufficiently in their conclusions. Specific comments: Introduction: I felt this addressed the key issues but was a little long Methods: Innovations reviewed were divided into groups, broadly categorised by "low tech" SMS/telephone, any internet content, or mixed. SMS/telephone just feels a bit mainstream now for most developed health care settings as much has been introduced on the assumption that it will work. Maybe this would have been better focussed if only studies from LMIDCs were included Quality assessment: I appreciate that the Newcastle-Ottawa guidelines were followed but could not see them reported anywhere - eg P8, line25/26: did all 6/6 studies show statistically significant results? the quality section, P9, reports pretty impressive levels of possible bias but this doesn't seem to have been integrated into the authors overall conclusions - see P10, line 33-41.
--

	I may well have missed the point with this review but am just struggling with its impressive, but I fear over-ambitious, breadth such that the overall assumption that online / digital is probably beneficial seems too bland to guide service planning and implementation especially as confidence in the greater part of the RCT & quasi-RCT literature on which it is based is acknowledged as subject to potential / real bias. The authors have done a great job of collating all the literature but I wonder if it would work better as a series of more focussed reviews.
--	---

REVIEWER	Francesco Sera Department of Social and Environmental Health Research, London School of Hygiene and Tropical Medicine, London, United Kingdom
REVIEW RETURNED	21-Jun-2017

GENERAL COMMENTS	This paper reports the results of a systematic review on interventions based on mobile phones or internet technologies, and their effects on outcomes related to HIV and sexually transmitted infections work. The authors choose to perform a comprehensive systematic research evaluating a broad spectrum of interventions, populations and health outcome. This was a challenge choice, but I think the authors were able to summarise the large amount of data collected, and to give a set of information that could be useful for public health policies and to plan future research. I have few minor comments on some aspects of the paper.  1. Following PRISMA figure 1 diagram I was surprise to see that non reference was identified through hand searching of the references. 2. I found Figure 2 difficult to follow (and to read). Would it be better having a table reporting the number of studies by technology (mhealth, internet, combined) in rows, and outcomes in columns? The proportion of significant results could be also reported in the columns. 3. The 99 studies exanimated have been collected in different continents. An acceptable number of studies have been conducted in North America, Europe, and Sub-Sharan Africa. It would been interesting to know if any difference in the interventions and its effects of the outcome was observed by geographical region. 4. The authors perform quantitative synthesis for a sub-group of papers evaluating clinic attendance as outcome and SMS reminders as intervention. One study Finocchiaro-Keller 2014 looks an outlier both in term of OR and precision. I would perform a sensitivity analysis excluding this study. The title of the related figure 3A need to be change with clear indication of the intervention, outcome and association measure considered. 5. The title of figure 3B looks wrong. It should contain a clear indication of the intervention, outcome and association measure considered. 6. I would avoid the report percentages when the denominator is low (e.g. 5 or under). For example line 54 page (1/1; 100%) would be simply (1/1).
--

VERSION 1 – AUTHOR RESPONSE

Reviewer: 1

Reviewer Name: Claudia Estcourt

Thank you for taking time to review our paper, we truly appreciate the comments and feedback.

Comment: It was good to see an assessment of quality of the literature reviewed but in the light of clearly documented issues of bias across the greater part of the RCTs & quasi RCTs included, I felt the authors did not address this in their abstract, nor discuss the possible implications of this sufficiently in their conclusions.

Response: Thank you. A statement on biases has been incorporated. A section on biases is highlighted in the discussion section. Please refer to page 2 lines 41-42 and page 11, lines 24-26.

Specific comments:

Comment: Introduction: I felt this addressed the key issues but was a little long

Response: Thank you. The introduction section has been edited to be concise. Please refer to page 4, lines 1-57.

Comment: Methods: Innovations reviewed were divided into groups, broadly categorised by "low tech" SMS/telephone, any internet content, or mixed. SMS/telephone just feels a bit mainstream now for most developed health care settings as much has been introduced on the assumption that it will work. Maybe this would have been better focused if only studies from LMIDCs were included

Response: Thank you. Allow us to explain. We wanted to make sure that the innovations were categorized well and proved for their acceptability.

Comment: Quality assessment: I appreciate that the Newcastle-Ottawa guidelines were followed but could not see them reported anywhere - eg P8, line25/26: did all 6/6 studies show statistically significant results? the quality section, P9, reports pretty impressive levels of possible bias but this doesn't seem to have been integrated into the authors overall conclusions - see P10, line 33-41

Response: Thank you. We have elaborated on biases in the discussion section (summary, conclusion and caveats section). We have included a note in the abstract. Please refer to page 2 lines 41-42 and page 11, lines 24-26.

Comment: I may well have missed the point with this review but am just struggling with its impressive, but I fear over-ambitious, breadth such that the overall assumption that online / digital is probably beneficial seems too bland to guide service planning and implementation especially as confidence in the greater part of the RCT & quasi-RCT literature on which it is based is acknowledged as subject to potential / real bias. The authors have done a great job of collating all the literature but I wonder if it would work better as a series of more focussed reviews.

Response: Please allow us to explain. A comprehensive review that provides a macroscopic picture of the field is an essential step towards guiding focused reviews. Isolated reviews exist but fail to provide the big picture that is often essential for service planning and impact initiatives. The field of innovations is moving at a fast pace and the field of research is not able to catch up with it. Our objective was to provide a gestalt of the field, with a note on what needs to be improved to make it more impactful.

Reviewer: 2

Reviewer Name: Francesco Sera

Thank you for taking time to review our paper, we truly appreciate the comments and feedback.

Comment: This paper reports the results of a systematic review on interventions based on mobile phones or internet technologies, and their effects on outcomes related to HIV and sexually transmitted infections work.

Response: Thank you for your comments.

Comment: The authors choose to perform a comprehensive systematic research evaluating a broad spectrum of interventions, populations and health outcome. This was a challenge choice, but I think the authors were able to summarise the large amount of data collected, and to give a set of information that could be useful for public health policies and to plan future research.

Response: Thank you!

Comment: Following PRISMA figure 1 diagram I was surprise to see that non reference was identified through hand searching of the references.

Response: Allow us to explain. Although hand-searching references are a very important method for compiling evidence, in our case this method identified articles that were already uncovered by our search strategy. Our search algorithm was tailored to four databases and yielded around 3000 de-duplicated records. We have conducted about 30 systematic reviews in the field of HIV/STI, and this method has worked well for us.

Comment: I found Figure 2 difficult to follow (and to read). Would it be better having a table reporting the number of studies by technology (mhealth, internet, combined) in rows, and outcomes in columns? The proportion of significant results could be also reported in the columns.

Response: Thank you. A table is included below as per your suggestion. This table is also a part of the appendices (Appendix 3). We have aimed to simplify Figure 2 and have uploaded the revised version.

Title: Table of studies by innovation (in rows) and by outcomes (in columns)

	Outcome Digital Innovation	Attendance rate	ART adherence	Risk reduction	Partner notification	Turnaround time	Self-care	Feasibility [†]	Acceptability [†]
Number of Studies by Type of Digital Innovation	mHealth Innovations (SMS/phone call only)	30*	24	6	0	2*	0	5	2
	Internet-based m/eHealth Innovations	6	4	5	0	0	1	4	1
	Combined innovations	1	1	0	2	0	1	3	1

Note: *1 study evaluated both attendance rate and turnaround time and was counted as part of the 30 studies on attendance rate. † studies reporting feasibility and acceptability as secondary outcomes are counted elsewhere in the table depending on primary outcome.

Comment: The 99 studies examined have been collected in different continents. An acceptable number of studies have been conducted in North America, Europe, and Sub-Saharan Africa. It would be interesting to know if any difference in the interventions and its effects of the outcome was observed by geographical region.

Response: Thank you for your suggestion. An analysis by geographical region on the three most reported outcomes (i.e. attendance rate, ART adherence, and risk reduction) revealed that studies conducted in North America reported the highest proportion of studies with positive effects. This is expected as the largest number of studies originated from North America. Within each continent, studies with effective results generally outweighed the ones that did not report an impact on outcomes; an exception was noted for ART adherence in Asia and risk reduction in Africa, where the total number of studies was small. Please find below a table illustrating this.

Title: Table of studies by geographic location and most commonly reported outcomes

Outcome Region	Attendance rate		ART Adherence		Risk reduction	
	Effective N (%)	No impact	Effective N (%)	No impact	Effective N (%)	No impact
North America	9 (82%)	2	8 (67%)	4	5 (56%)	4
South America	1	-	-	1	-	-
Europe	7 (67%)	4	3	-	-	-
Africa	7 (70%)	3	5 (63%)	3	1 (50%)	1
Asia	-	1	1 (33%)	2	-	-
Australia	3	-	2	-	-	-
TOTAL	27	10	19	10	6	5

Comment: The authors perform quantitative synthesis for a sub-group of papers evaluating clinic attendance as outcome and SMS reminders as intervention. One study Finocchiaro-Keller 2014 looks an outlier both in term of OR and precision. I would perform a sensitivity analysis excluding this study. The title of the related figure 3A need to be change with clear indication of the intervention, outcome and association measure considered.

Response: Thank you for your suggestion; it is indeed a very interesting analysis to perform. After excluding the outlier study by Finocchiaro-Keller we obtained the following revised pooled OR= 1.69 [95%CI: 1.23, 2.33]. The difference in summary statistic is relatively small compared to our original result (pooled OR=1.76 [95%CI: 1.28, 2.42]). Hence, the sensitivity analysis validates our initial conclusion. We have included the sensitivity analysis below. The title of Figure 3A was “Sub-Group Analysis Pooled OR for Attendance” and should be indicated on the uploaded figure.

Sensitivity Analysis Pooled OR for Attendance

Comment: The title of figure 3B looks wrong. It should contain a clear indication of the intervention, outcome and association measure considered.

Response: Thank you for bringing our attention to this. The title of Figure 3B should have actually been “Sub-Group Analysis Pooled OR for Adherence” this should be indicated on the uploaded figure.

Comment: I would avoid the report percentages when the denominator is low (e.g. 5 or under). For example line 54 page (1/1; 100%) would be simply (1/1).

Response: Thank you, we have addressed this as per your suggestion and eliminated percentage values for denominators 5 or under in both the abstract section and results section, found on page 2 lines 31-42 and page 8 lines 33-54, respectively.

VERSION 2 – REVIEW

REVIEWER	Francesco Sera Department of Social and Environmental Health Research, London School of Hygiene & Tropical Medicine, London, United Kingdom. None
REVIEW RETURNED	29-Aug-2017

GENERAL COMMENTS	I read the revised version of this systematic review on interventions based on mobile phones or internet technologies, and their effects on outcomes related to HIV and sexually transmitted infections, and I confirm my thought that the main quality of this study is the set of information systematically collected and synthesised by the authors. This set of information, in my point of view, could be useful for public health policies and to plan future research. The authors answered positively to the point I rose in my first review. There are two minor points that can be further addressed: 1. In Figure 2 the labels of the categories are still very small. Also the labels explaining the meaning for the different colours (interventions) are unreadable.2. Page 6. Lines 8. I think “Dersimonian” is “DerSimonian”. I would remove “weighted by study sample”. In fact the set of weight is dependent by the standard error (function of the sample size) and the between-study variability estimated by the methods of moments. A more correct statement would be “we used the random effect meta-analysis model with DerSimonian and Lairs estimator (moments method) of the between-study variance to calculate the pooled effect”.
--

VERSION 2 – AUTHOR RESPONSE

We included the reviewers suggested write up. Please refer to the tracked version, on page 6, paragraph 1, lines 5 and 6.

We enlarged the font size for figure 2.

We hope it enhances readability. We have uploaded a tracked changed version and a clean version.